# Chrysomycin A Regulates Proliferation and Apoptosis of Neuroglioma Cells via the Akt/GSK-3β Signaling Pathway In Vivo and In Vitro

**DOI:** 10.3390/md21060329

**Published:** 2023-05-27

**Authors:** Dong-Ni Liu, Man Liu, Shan-Shan Zhang, Yu-Fu Shang, Wen-Fang Zhang, Fu-Hang Song, Hua-Wei Zhang, Guan-Hua Du, Yue-Hua Wang

**Affiliations:** 1Beijing Key Laboratory of Drug Target Identification and New Drug Screening, Institute of Materia Medica, Chinese Academy of Medical Sciences & Peking Union Medical College, Beijing 100050, China; liudongni@imm.ac.cn (D.-N.L.); liuman@imm.ac.cn (M.L.); zhangshanshan@imm.ac.cn (S.-S.Z.); shangyufu@imm.ac.cn (Y.-F.S.); zhangwenfang@imm.ac.cn (W.-F.Z.); 2School of Light Industry, Beijing Technology and Business University, Beijing 100048, China; songfuhang@btbu.edu.cn; 3School of Pharmaceutical Sciences, Zhejiang University of Technology, Hangzhou 310014, China; hwzhang@zjut.edu.cn

**Keywords:** Chrysomycin A, xenograft mouse model, glioblastoma, apoptosis

## Abstract

Glioblastoma (GBM) is a major type of primary brain tumor without ideal prognosis and it is therefore necessary to develop a novel compound possessing therapeutic effects. Chrysomycin A (Chr-A) has been reported to inhibit the proliferation, migration and invasion of U251 and U87-MG cells through the Akt/GSK-3β signaling pathway, but the mechanism of Chr-A against glioblastoma in vivo and whether Chr-A modulates the apoptosis of neuroglioma cells is unclear. The present study aims to elucidate the potential of Chr-A against glioblastoma in vivo and how Chr-A modulates the apoptosis of neuroglioma cells. Briefly, the anti-glioblastoma activity was assessed in human glioma U87 xenografted hairless mice. Chr-A-related targets were identified via RNA-sequencing. Apoptotic ratio and caspase 3/7 activity of U251 and U87-MG cells were assayed via flow cytometry. Apoptosis-related proteins and possible molecular mechanisms were validated via Western blotting. The results showed that Chr-A treatment significantly inhibits glioblastoma progression in xenografted hairless mice, and enrichment analysis suggested that apoptosis, PI3K-Akt and Wnt signaling pathways were involved in the possible mechanisms. Chr-A increased the apoptotic ratio and the activity of caspase 3/7 in U251 and U87-MG cells. Western blotting revealed that Chr-A disturbed the balance between Bax and Bcl-2, activating a caspase cascade reaction and downregulating the expression of p-Akt and p-GSK-3β, suggesting that Chr-A may contribute to glioblastoma regression modulating in the Akt/GSK-3β signaling pathway to promote apoptosis of neuroglioma cells in vivo and in vitro. Therefore, Chr-A may hold therapeutic promise for glioblastoma.

## 1. Introduction

Glioblastoma (GBM), currently accounting for 48.6% of malignant brain tumors, is a major type of primary brain tumor, causing high levels of necrosis, endothelial cell proliferation, invasion and angiogenesis. GBM belongs to the highest level (Level 4) in the classification of brain tumors by the World Health Organization (WHO) [1,2,3,4]. Surgery, radiotherapy and chemotherapy are classic therapeutic methods for GBM patients. However, these methods can only improve the survival duration and survival rate of patients to a limited extent [5]. Poor prognosis and significant neurologic morbidity of GBM, especially among the elderly, reflects a high social and medical burden [4,6]. Temozolomide (TMZ), lomustine and carmustine are commonly approved chemotherapeutic drugs, but these are mostly followed by modest recurrence benefits [7]. The five-year survival rate has only improved from 4% to 7%, with even less breakthroughs in the prevention and diagnosis of GBM patients in the past 40 years [4]. Therefore, there is an urgent need to identify new effective therapeutic options for glioblastoma.

Chrysomycin A (Chr-A, Figure 1A), a type of glycoside with benzonaphthopyranone, was first discovered in 1955 [8,9]. In recent years, progress has been made in the production conditions and preparation of Chr-A, which is derived from *Streptomyces* sp. *891* marine sediment, have contributed to its new drug development processes [10,11]. It has been reported that Chr-A has antibiotic, anti-tumor, anti-tuberculosis, and anti-neuroinflammation properties [12,13,14]. The inhibitory effect and possible mechanisms of Chr-A on proliferation, migration and invasion towards U251 and U87 glioblastoma cells has been investigated previously [15]. However, there are no previous reports of Chr-A exerting anti-glioblastoma effects in vivo. Thus, in this study we evaluate its molecular mechanisms and regulatory biological processes in human glioma U87 xenografted hairless mice. 

Apoptosis, a programmed cell death initiated under physiological and pathological conditions, serves as a key entry point for many therapeutic strategies against cancer [16,17]. In tumor cells, apoptosis is usually inhibited, allowing indefinite proliferation of malignant tumor cells. Among the complex mechanisms regulating apoptosis, Bcl-2 family members, P53 and caspases are novel targets for therapies towards treating multiple cancers [18]. 

Herein, we investigated the regulation and possible mechanisms of Chr-A on the apoptosis of human glioblastoma cells in human glioma U87 xenografted hairless mice, assisted by RNA-sequencing. We further explore and validate how Chr-A mediates the apoptotic ratio and apoptosis-related caspase cascade in human U87 and U251 cells. Our study of Chr-A provides valuable evidence towards the treatment of glioblastoma, as well as towards the developmental prospects of other marine-derived drugs.

## 2. Results

### 2.1. Chr-A Suppressed the Tumorigenicity in Human Glioma U87 Xenografted Hairless Mice

To evaluate the anti-tumor potential of Chr-A in vivo, we established a xenograft model by inoculating hairless mice with U87-MG cells and 3 mg or 10 mg/kg Chr-A was administered via intraperitoneal injection once per day for 18 days. During Chr-A treatment, no obvious weight loss or abnormal behavior was observed (Figure 1B). The tumor weight and tumor volume were both significantly decreased following administration of Chr-A (Figure 1C–E). Furthermore, no mortality occurred, nor were any significant changes in the colors and textures of vital organs observed, including the liver, kidney, heart, lung and spleen, and no significant differences in the relative organ weights were observed in the Chr-A treatment groups compared to the vehicle group (Figure 1F). In addition, to determine the inhibitory effects of Chr-A on proliferation of glioblastoma cells in hairless mice, Ki67 immunofluorescence staining was performed. The number of Ki67 positive cells (red fluorescent) of Chr-A treatment groups, which indicated the marker of cell proliferation, decreased significantly compared to the vehicle group (Figure 1G,H). Thus, Chr-A played an inhibitory role towards the viability and proliferation of human glioblastoma cells, leading to tumor regression of xenografted hairless mice.

### 2.2. Chr-A Regulates Akt/GSK-3β Signaling Pathway of Glioblastoma Cells in Human Glioma U87 Xenografted Hairless Mice

To better understand the underlying mechanisms of Chr-A against GBM, an RNA-sequence analysis of tumor tissues of BALB/c hairless mice was performed first, and there were 992 differential expression genes (DEGs) regulated by Chr-A (Figure 2A,B, Appendix A). In addition, a total of 5299 genes were obtained by searching the GeneCard, DrugBank Online and Online Mendelian Inheritance in Man (OMIM) databases (Appendix A). Further analysis showed there were 383 overlapped genes marking Chr-A-related targets and glioblastoma-related targets, which were regarded as potential therapeutic targets of Chr-A against glioblastoma in the present study (Figure 2C, Appendix A). Subsequently, the underlying pharmacological mechanisms of Chr-A against glioblastoma could be enlightened by GO and KEGG enrichment analyses. The top 20 significant GO terms, including biological process (BP), cellular component (CC) and molecular function (MF), are shown in Figure 2D–F. The significant signaling pathways identified by KEGG enrichment analysis were mainly involved in the PI3K-Akt signaling pathway, Wnt signaling pathway, Apoptosis, and so on (Figure 2G). Abnormal regulation of PI3K-Akt and Wnt signaling pathways, which work together via the communication between Akt and GSK-3β, facilitates GBM progression. Our results showed that Chr-A significantly downregulated Akt, p-Akt, and p-GSK-3β of glioblastoma cells in hairless mice and influenced the expression of GSK-3β with no significant difference (Figure 3A,B). Furthermore, Chr-A remarkably reduced the expression of slug and MMP2, while the downstream of Wnt signaling pathway resulted from Akt/GSK-3β signal modulation (Figure 3C,D). Thus, Chr-A may exert anti-glioblastoma activity in vivo via the Akt/GSK-3β signaling pathway.

### 2.3. Chr-A Induces Apoptosis of U251 and U87-MG Cells

Annexin Ⅴ/PI staining was used to detect the percent of living cells (Q4), early apoptotic cells (Q3), late apoptotic cells (Q2), and necrotic cells (Q1) of glioblastoma cells after 48h of Chr-A treatment. Compared to the control group, the apoptosis rate (Q3 + Q2) of U251 and U87-MG cells remarkably increased in Chr-A groups as the concentration increased (Figure 4A,B). Caspase 3 and Caspase 7 are important executive factors for cell apoptosis, whose activity reflects the occurrence of cell apoptosis to a certain extent. The results showed that Chr-A significantly increased caspase 3/7 activity in U251 and U87-MG cells with different concentrations of Chr-A compared with control group (Figure 4C,D).

### 2.4. Chr-A Increases Ratio of Bax to Bcl-2 of Glioblastoma Cells In Vivo and In Vitro

Members of Bcl-2 family proteins are indispensable for apoptosis, among which Bax functions as a pro-apoptotic protein and Bcl-2 serves as anti-apoptotic protein. To maintain physiological homeostasis, Bax and Bcl-2 maintain a balance regulating cell metabolism. Western blotting showed that Chr-A treatment caused a remarkably elevated ratio of Bax/Bcl-2 neuroglioma cells in xenografted hairless mice (Figure 5A,B). Meanwhile, compared to the control group, Chr-A treatment for 48 h significantly heightened the ratio of Bax/Bcl-2 in U251 cells and U87-MG cells (Figure 5C–E). Together, these results indicated that Chr-A regulates the balance between Bax and Bcl-2, inducing apoptosis of glioblastoma cells in vivo and in vitro.

### 2.5. Chr-A Activates a Caspase Cascade Reaction of Glioblastoma Cells In Vivo and In Vitro 

Apoptosis is mainly regulated in two pathways: the mitochondrial apoptosis pathway and the death receptor apoptosis pathway, both of which cannot come into play without the activation of a caspase cascade reaction, including caspase 3 and caspase 7. The results showed that Chr-A treatment caused a remarkably elevated ratio of caspase 9/caspase 9, as well as an elevated expression ratio of cleaved caspase 3/caspase 3, cleaved caspase 7/caspase 7 in glioblastoma cells of tumor tissues, which are executives downstream of the apoptotic process (Figure 6A,B). The same regulation was also detected on U251 and U87-MG cells with Chr-A treatment for 48 h in vitro (Figure 6C–E). In summary, our data suggested that the inhibition of glioblastoma by Chr-A is due to activation of caspases leading to apoptosis.

## 3. Discussion

Glioblastoma, characterized by great aggressiveness, poor prognosis and short-term survival, is the primary and most lethal tumor in the central nervous system. Surgery, radiotherapy and chemotherapy were widely adopted therapeutic approaches to glioblastoma but the survival rates of patients were not improved ideally [2,7]. TMZ was an oral chemotherapy commonly used for patients with glioblastoma containing nonnegligible adverse effects [19], and among the patients, 55% showed resistance to TMZ [20]. Thus, developing effective therapeutics for the treatment of glioblastoma is of the utmost importance.

Chr-A, a rarely studied marine drug, has a group of benzonaphthopyranone glycosides, which are suggested to possess antitumor activity [9]. Pharmacokinetic evaluations of Chr-A showed that the brain was one of the major tissues with distribution in female ICR mice after an oral dose of 50 mg/kg of Chr-A, suggesting that Chr-A has the ability to cross the blood-brain-barrier and enter the brain. To improve solubility, researchers prepared Chr-A mixed with disodium glycyrrhizin to promote self-micelle solid dispersion and showed improved solubility and enhanced oral bioavailability in terms of the area under the plasm concentration–time curve and the half-life but not the peak plasm concentration or time to reach peak concentration [10]. Previous studies determined that Chr-A showed inhibitory effects towards human U251 and U87-MG glioblastoma cells in vitro via the Akt/GSK-3β signaling pathway [15]. In the current study, we verified the antitumor activity of Chr-A in vivo using human U87-MG cells xenografted into glioblastoma models, from which we discovered that intraperitoneal administration of both 3 mg and 10 mg/kg Chr-A suppressed tumorigenicity. Importantly, there is no obvious change in the weight, behavior and organs of mice during administration, thus indicating that Chr-A has an effect against GBM with low toxicity to mice as other published reports have concluded [9]. Furthermore, Chr-A has no influence on the lysis of red blood cells [21]. To explore the in-depth molecular characterization of signaling pathways, we conducted an RNA-sequence analysis of tumor tissue obtained from xenografted glioblastoma in hairless mice. 383 intersection genes between differentially expressed genes screened from the RNA-sequence analysis and glioblastoma-related targets collected from various databases were regarded as the core genes for Chr-A against glioblastoma in the study. According to the KEGG enrichment analysis, we finally focused on apoptosis, PI3K-AKT and Wnt signaling pathways to explore the mechanism that Chr-A uses against glioblastoma. 

Apoptosis is an ordered physiological process taking place in cells under pathological conditions. Targeting apoptosis-related proteins plays an important role in medicines exerting certain therapeutic effects on glioblastoma [22,23]. It is the altered expression ratio of multiple pro-and anti-apoptotic proteins, rather than the absolute quantity of those proteins, that modulates cell death. Bcl-2 family proteins are indispensable in the apoptotic process. Bax, a member of Bcl-2 family, can accelerate cytochromes and release them into the cytoplasm, resulting in activating Caspase-3 during apoptosis promotion. Bcl-2, an anti-apoptosis protein, favors intracellular Ca^2+^ release, as it encodes mitochondrial outer membrane proteins [24,25]. Mitochondrial membrane permeability becomes impaired when the Bax/Bcl-2 ratio becomes unbalanced, followed by Cyt C and AIF releases and caspase-9 activation, furthering caspase-3/7 activation [26]. As shown in our results, Chr-A treatment stimulated the apoptosis heightening ratio of Bax/Bcl-2 and activated caspases, including caspase 3, caspase 7 and caspase 9 of glioblastoma cells in vivo and in vitro, thus confirming the enrichment analysis that Chr-A leads to tumor regression via apoptosis-promoting.

Abnormal activation of the PI3K-Akt and Wnt signaling pathways is closely related to cytoskeletal rearrangement, metabolism, apoptosis, and angiogenesis in GBM [27,28]. Interestingly, the interaction between Akt and GSK-3β builds a bridge between the PI3K-Akt and Wnt signaling pathways to make it possible to regulate oncogenesis synergistically: Akt can not only recognize the accumulation of phosphatidylinositol 3,4,5-trisphosphate (PIP3) provoked by PI3K to stimulate a downstream of the classic PI3K-Akt signal [29], but also the accumulation of phosphorylate GSK-3β at the Ser9 site [30], further regulating β-catenin localization in Wnt signals and therefore affecting other inhibitors downstream, including slug and MMP2, to influence tumor growth [31,32]. Akt inhibitors can inhibit growth and induce the apoptosis of glioblastoma cells through decreasing the level of β-catenin nuclear translocations [33]. In our study, a remarkable downregulation of p-Akt and p-GSK-3β in the Chr-A group was observed following Chr-A treatment, and their downstream, slug and MMP2, were also shown to be significantly decreased. Our previous study in vitro has already shown that Chr-A could downregulate Akt/GSK-3β signaling pathways in U87 and U251 cells [15]. Thus, Chr-A may function against glioblastoma via apoptosis regulation in the Akt/GSK-3β signaling pathways in vivo and in vitro, confirming the KEGG enrichment analysis above.

## 4. Materials and Methods

### 4.1. Reagents

Chr-A was provided by Prof. Hua-Wei Zhang (Zhejiang University of Technology). Dulbecco’s modified Eagle’s medium (DMEM) and fetal bovine serum (FBS, Cat# 164210-50) for cell cultures were purchased from Gibco BRL (Grand Island, NY, USA) and Procell (Wuhan, China), respectively. Anti-Bax antibodies (50599-2-Ig) and anti-Bcl-2 antibodies (12789-1-AP) were purchased from Proteintech (Manchester, United Kingdom). Anti-Caspase 3 antibodies (9662), anti-Cleaved Caspase 3 antibodies (9664), anti-Caspase 7 antibodies (12827), anti-Cleaved Caspase 7 antibodies (8438), anti-Caspase 9 antibodies (9508), anti-Cleaved Caspase 9 antibodies (20750), anti-Akt antibodies (9272), anti-p-Akt antibodies (9271), anti-GSK-3β antibodies (9315), anti-p-GSK-3β antibodies (9323), anti-slug antibodies (9585), anti-MMP2 antibodies (87809) and anti-GAPDH antibodies (5174) were purchased from Cell Signaling Technology (Beverley, CA, USA).

### 4.2. Antitumor Activity in Xenografted Tumor Mouse Models

Female BALB/c hairless mice (6–7 weeks) were purchased from Beijing Vital River Laboratory Animal Technology Co., Ltd. (Beijing, China; animal certification number SCXK(Jing) 2021–0006), weighing 14–17 g. All animal care and experimental procedures regarding the animals were approved by the ethics committees of the Institute of Materia Medica, the Chinese Academy of Medical Sciences, and Peking Union Medical College; approval number 00005431. Briefly, U87-MG glioblastoma cells in the logarithmic phase (1 × 10^7^ cells per mouse) were subcutaneously implanted into the right flanks of the mice. When the tumor volume was approximately 100–150 mm^3^, the mice were randomly divided into three groups (*n* = 9) and then injected intraperitoneally with vehicle (normal saline with 0.1% Tween 80) or with Chr-A (3 mg or 10 mg/kg) once per day for another 18 days. The dosage of Chr-A was based on our previous study. The tumor volume was measured via a Vernier caliper and calculated using the following formula: tumor volume (mm^3^) = 0.5 × (length) × (width)^2^. At the conclusion of the experiment, all mice were euthanized. The tumor tissues were collected and the tumor weights and organ weights were recorded. 

### 4.3. Ki67 Staining via Immunofluorescence

The expression of Ki67 in tumor tissues was assayed by immunofluorescence, which was detected by Wuhan Servicebio technology (Wuhan, China). Briefly, tumor tissues were fixed in 4% PFA overnight at 4 °C and embedded in paraffin. Next, tumor tissue sections were incubated with anti-Ki67 primary antibodies ((Servicebio, GB111141, 1: 200) at 4 °C overnight after blocking with BSA (Servicebio, G5001) for 30 min and permeabilization. After washing with PBS, the sections were incubated with Cy3 labeled secondary antibodies at room temperature for 2 h. Nuclei were stained with 4,6-diamidino-2-phenylindole (DAPI) for 10 min at room temperature. All images were acquired using a fluorescence microscope and fluorescence intensity was analyzed using the image processing package Image J/Fiji.

### 4.4. Chr-A-Related Target Identification via RNA-Sequencing

RNA-sequencing was detected by Shanghai Kangcheng Biology (Shanghai, China). The brief methods are as follows: the total RNA of U87-MG xenografts in the hairless mice was extracted and sequenced by an Illumina NovaSeq 6000 Sequencer (San Diego, CA, USA). The image processing and base classification were carried out by a Solexa pipeline version 1.8 (offline base caller software, version 1.8, San Diego, CA, USA), and the differential genes regulated by Chr-A were further analyzed and screened [34,35].

### 4.5. Glioblastoma-Related Target Collection and Analysis

GeneCard (https://www.genecards.org/, accessed on 23 November 2021), DrugBank Online (https://go.drugbank.com/, accessed on 23 November 2021) and the Online Mendelian Inheritance in Man (https://omim.org/, accessed on 23 November 2021) were used to obtain glioblastoma-related targets by searching “glioblastoma” on the platform. After discarding the duplicate targets, the glioblastoma-related genes were described. The Metascape database (https://metascape.org/, accessed on 23 November 2021) was used for GO and KEGG enrichment analyses. Metascape is an online tool for gene annotation and analysis resources. It is often used to process enrichment analysis of Gene Ontology (GO) and Kyoto Encyclopedia of Genes and Genomes (KEGG) pathways.

### 4.6. Glioblastoma Cell Culture and Treatment

U251 and U87-MG cells were obtained from the Cell Bank of the Chinese Academy of Sciences (Beijing, China). Cells were cultured with DMEM containing 10% FBS in an incubator at 37 ℃ constantly maintaining a humidified atmosphere of 5% CO2. Chr-A was added to U87 cells at 0, 0.2, 0.6, 1.8 μM and added to U251 cells at 0, 0.2, 0.4, 0.8 μM according to our previous study [15]. 

### 4.7. Apoptosis Assay

A TransDetect® Annexin V-EGFP/PI Cell Apoptosis Detection Kit (TransGen Biotech, Beijing, China) was used to determine the percent of viable cells, early and late apoptotic cells and necrotic cells by flow cytometry (BD FACSVerse, San Diego, CA, USA). 1.5 × 10^5^ cells were seeded in each well of the 6-well plates for 24 h before replacing the culture media with serum-free DMEM, and adding 0, 0.2, 0.4, and 0.8 μM Chr-A to U251 cells; 0, 0.2, 0.6, and 1.8 μM Chr-A were added to U87-MG cells for 48 h. Next, cells were harvested and processed according to the detection kit instruction manual. Detections were carried out via flowcytometry and the data was analyzed using Flow Jo software (Tristar, CA, USA).

### 4.8. Caspase 3/7 Activity Detection

A Caspase-3/7 Live-cell Fluorescence Real-time Detection Kit (KeyGenBioTECH, Nanjing, China) was used to assess caspase-3/7 activity. Cells were seeded in 6-well plates for 24 h before replacing the culture media with serum-free DMEM and adding 0, 0.2, 0.4, and 0.8 μM Chr-A to U251 cells; 0, 0.2, 0.6, and 1.8 μM Chr-A were added to U87-MG cells for 48 h. Next,, cells were harvested, washed with PBS, and the Caspase 3/7 activity was determined according to the instruction manual of the detection reagent. Detections were carried out via flow cytometry (BD Accuri C6, San Diego, CA, USA).

### 4.9. Western Blotting

The total proteins of glioblastoma cells and tissues were extracted with a RIPA lysis buffer (ApplyGen, Beijing, China) in an ice water bath. After centrifugation at 12,000 rpm for 15 min, quantification with a BCA Protein Assay Kit (ApplyGen, Beijing, China) was performed. Subsequently, the proteins were separated by a 10% SDS-PAGE gel and transferred to polyvinylidene difluoride membranes (Millipore, Billerica, MA, USA). Next, membranes were blocked by 5% skim milk in TBST for 2 h at room temperature, followed by an incubation with primary antibodies at 4 ℃ overnight. Subsequently, the membranes were incubated with horseradish peroxidase-conjugated secondary antibodies for 1 h at room temperature. An ECL hypersensitive luminescence solution was then used to visualize the protein bands. The grayscale value of the bands on the images were analyzed by Image J software (Version 2., National Institutes of Health, Bethesda, USA).

### 4.10. Statistical Analysis

A statistical analysis was performed using the GraphPad prism7 (Version7.0., GraphPad Software, San Diego, CA, USA). The results were expressed as mean ± SD. The differences in the groups were analyzed by one-way ANOVA Multiple comparisons. *P* < 0.05 was considered statistically significant.

## 5. Conclusions

Our study has shown that Chr-A has the potential to suppress oncogenesis of glioblastoma by inducing apoptosis through the Akt/GSK-3β signaling pathway. However, there are still more aspects that need to be explored and verified as implied by the enrichment analysis, such as cell cycle arrests of the glioblastoma cells and other underlying mechanisms. Next, we may further study the therapeutic mechanisms of Chr-A on glioblastoma and hope to identify some specific targets of Chr-A for the treatment of glioblastoma.

## Figures and Tables

**Figure 1 marinedrugs-21-00329-f001:**
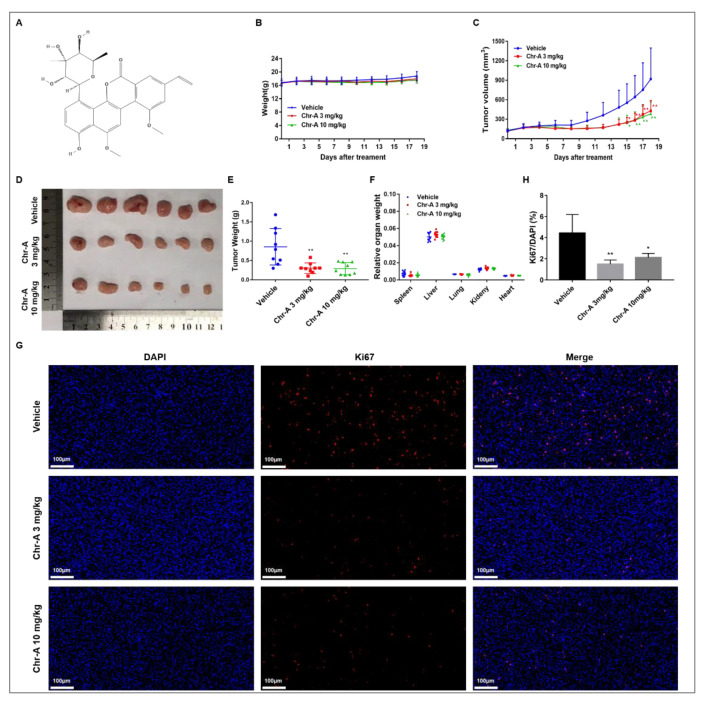
Chr-A promotes tumor regression and suppresses proliferation of glioblastoma cells in xenografted hairless mice. (**A**) Molecular formula of Chr-A, (**B**) Changes in body weight during the Chr-A administration period, (**C**) Changes in tumor volumes during the Chr-A administration period, (**D**) Xenograft tumors, (**E**) Tumor weights at the conclusion of the experiment, (**F**) Relative organ weights at the conclusion of the experiment, *n* = 9; (**G**) Representative fluorescent micrographs of immunofluorescent staining with Ki67 antibodies, (**H**) Statistical analysis of the percentage of Ki67 positive cells, white scale bars represent 100 μm, *n* = 3. The data were presented as mean ± SD, * *P* < 0.05, ** *P* < 0.01 vs. vehicle group.

**Figure 2 marinedrugs-21-00329-f002:**
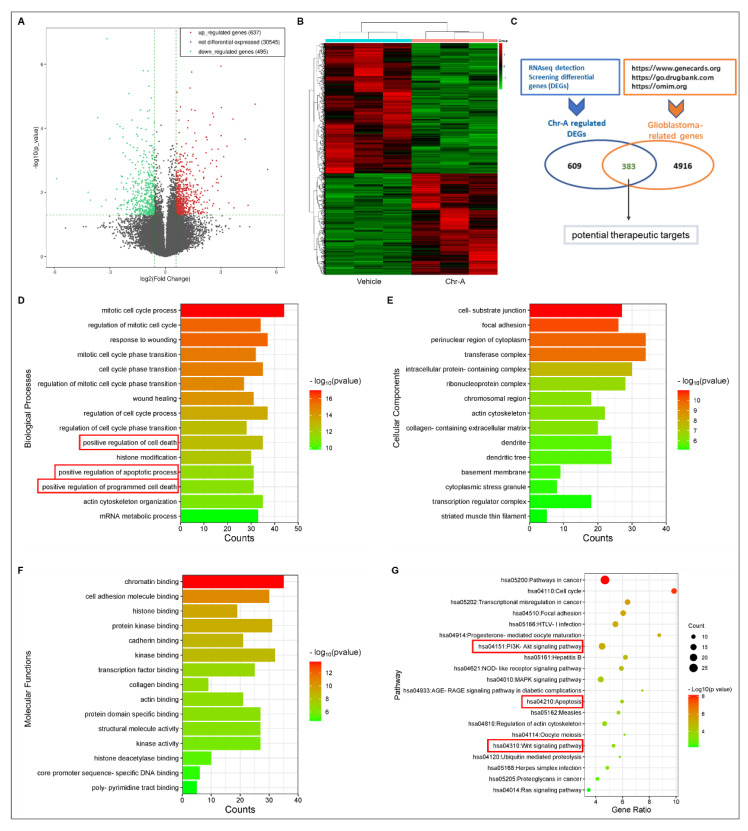
Analysis of potential targets for Chr-A against glioblastoma. (**A**) Volcano map of differential genes (up-regulated genes are in red; down-regulated genes are in green; fold change > 1.5, *P* value < 0.05); (**B**) Heat map of differential gens; (**C**) Venn diagram of targets; (**D**) Top 15 significant biological processes (BP) terms, terms of interest were highlighted wirh red box; (**E**) Top 15 significant cellular components (CC) terms; (**F**) Top 15 significant molecular functions (MF) terms; (**G**) Top 20 significant KEGG pathways, terms of interest were highlighted wirh red box.

**Figure 3 marinedrugs-21-00329-f003:**
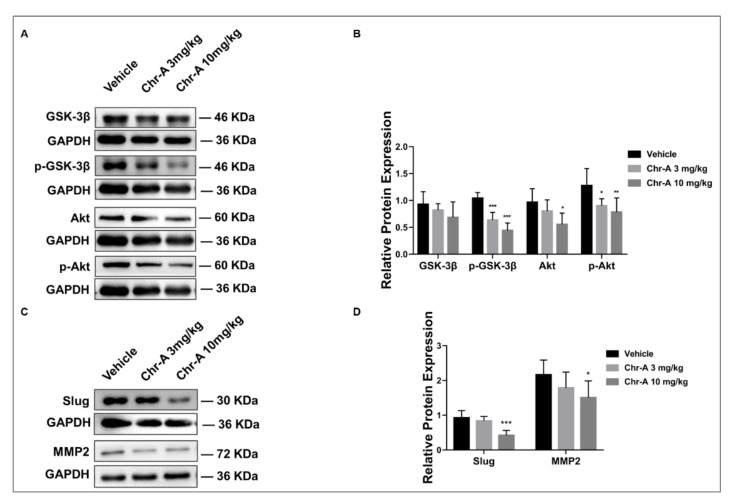
Validation of possible mechanisms for Chr-A against glioblastoma in vivo. (**A**,**B**) Chr-A decreased expression of GSK-3β, p-GSK-3β, Akt, and p-Akt of glioblastoma cells in tumor tissues, (**C**,**D**) Chr-A downregulated slug and MMP2 of glioblastoma cells in tumor tissues. The data were presented as mean ± SD (*n* = 5–6), * *P* < 0.05, ** *P* < 0.01, *** *P* < 0.001 vs. vehicle group.

**Figure 4 marinedrugs-21-00329-f004:**
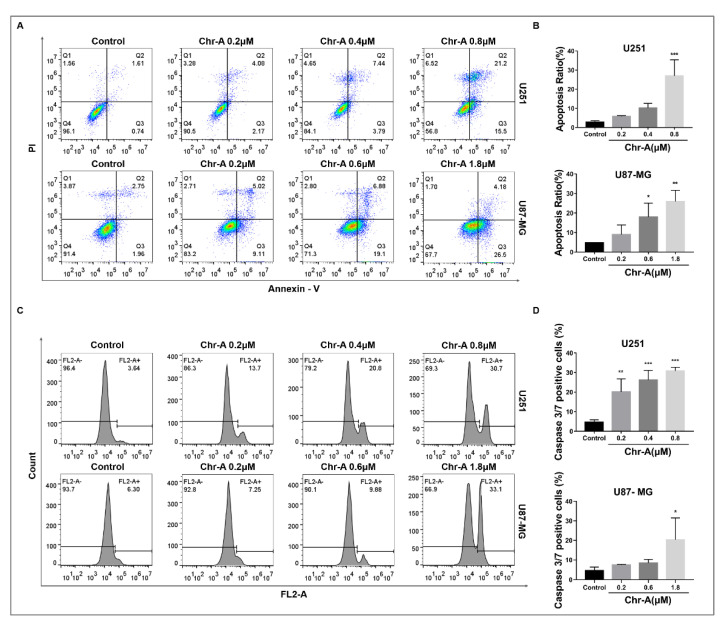
Chr-A increases the apoptosis rate and caspase 3/7 activity in U251 and U87-MG cells. (**A**,**B**) Chr-A increased the apoptosis rate of U251 and U87-MG cells, (**C**,**D**) Chr-A activated caspase 3/7 activity of U251 and U87-MG cells. The data were presented as mean ± SD (*n* = 3), * *P* < 0.05, ** *P* < 0.01, *** *P* < 0.001 vs. vehicle group.

**Figure 5 marinedrugs-21-00329-f005:**
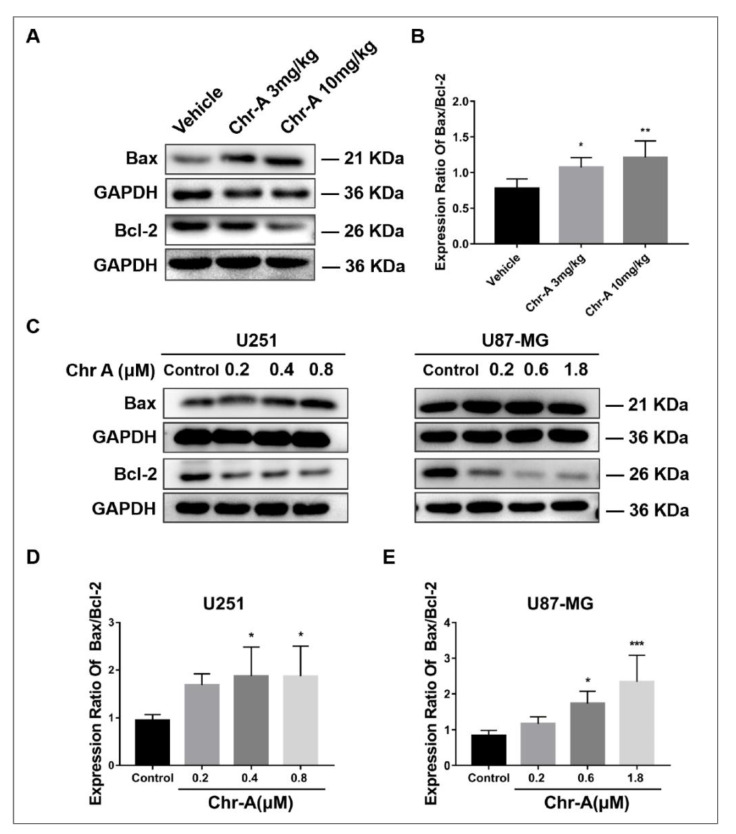
Chr-A increases the expression ratio of Bax to Bcl-2 to regulate apoptosis in vivo and in vitro. (**A**,**B**) Chr-A increases the expression ratio of Bax to Bcl-2 in glioblastoma cells of tumor tissues, *n* = 6, (**C**–**E**) Chr-A increases the expression ratio of Bax to Bcl-2 in U251 and U87-MG cells, *n* = 4. The data were presented as mean ± SD, * *P* < 0.05, ** *P* < 0.01, *** *P* < 0.001 vs. the vehicle group and the control group.

**Figure 6 marinedrugs-21-00329-f006:**
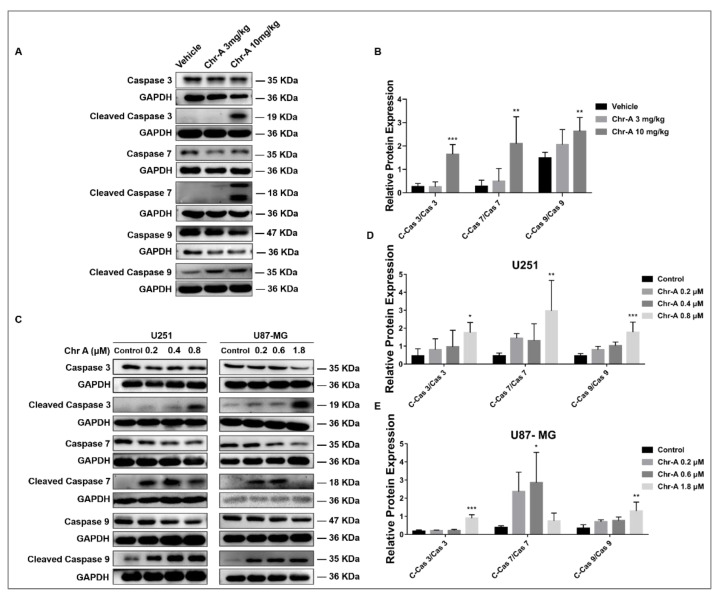
Chr-A activates caspases 3/ 7/ 9 to mediate the apoptosis of glioblastoma cells in vivo and in vitro. (**A**,**B**) Chr-A increases the expression ratio of cleaved caspase3/ caspase3(C-Cas3/Cas3), cleaved caspase7/ caspase7(C-Cas7/Cas7), cleaved caspase9/ caspase9(C-Cas9/Cas9) of glioblastoma cells in tumor tissues, *n* = 6, (**C**–**E**) Chr-A increases the expression ratio of cleaved caspase3/ caspase3(C-Cas3/Cas3), cleaved caspase7/ caspase7(C-Cas7/Cas7), cleaved caspase9/ caspase9(C-Cas9/Cas9) in U251 and U87-MG cells, *n* = 3–4.The data were presented as mean ± SD, * *P* < 0.05, ** *P* < 0.01, *** *P* < 0.001 vs. the vehicle group and the control group.

## Data Availability

The original data presented in the study are included in the article/Appendix A; further inquiries can be directed to the corresponding author.

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
