# Peer review of "Chrysomycin A Regulates Proliferation and Apoptosis of Neuroglioma Cells via the Akt/GSK-3β Signaling Pathway In Vivo and In Vitro"

_marinedrugs, 2023, doi:10.3390/md21060329_

Round 1

Reviewer 1 Report

The interest in finding new therapeutic strategies and/or new targets is still very high in glioblastoma research. This is because the efforts made up to now have certainly led to a better knowledge of the biology of this tumor but have not allowed a great improvement in the patient's prognosis. Therefore, identifying new therapeutic approaches, alternative to the conventional ones, is certainly innovative and of interest.

Thus, the manuscript is novel and interesting. It is the continuation of the work previously published by the same authors. However, some point should be improved to better clarify.

 -The in vivo experimental model is a heterotopic xenograft. The administration of Chr-A is performed by subcutaneous injection. What about the ability of this compound to cross the blood brain barrier? Have the authors information about this point? Are there other studies that evaluated the drug bioavailability? In any case, it would be appropriate to insert a consideration on this point in the Discussion section.

 -Authors showed that Chr-A induces apoptosis of glioma cells both in vivo and in vitro by investigating cleaved-caspase expression levels with western blot technique. to complete better, it would be interesting to evaluate the apoptotic rate in the glioma tissue samples of the control and treated group. In addition, given that data suggest the activation of the mitochondrial pathway, authors should investigate, in vitro, the mitochondria status (for example by using flow cytometry and JC-1 staining) in untreated or Chr-A-treated glioma cells.

 -Figure 1 panel G and Figure 2 especially panels A, B and C are too small. Impossible to appreciate the data. Please improve the size and resolution.

 -The size of the lettering on figures 3 and 5 is too small. Please improve.

 Author Response

Response to Reviewer 1

The interest in finding new therapeutic strategies and/or new targets is still very high in glioblastoma research. This is because the efforts made up to now have certainly led to a better knowledge of the biology of this tumor but have not allowed a great improvement in the patient's prognosis. Therefore, identifying new therapeutic approaches, alternative to the conventional ones, is certainly innovative and of interest. Thus, the manuscript is novel and interesting. It is the continuation of the work previously published by the same authors. However, some point should be improved to better clarify.

Point 1 The in vivo experimental model is a heterotopic xenograft. The administration of Chr-A is performed by subcutaneous injection. What about the ability of this compound to cross the blood brain barrier? Have the authors information about this point? Are there other studies that evaluated the drug bioavailability? In any case, it would be appropriate to insert a consideration on this point in the Discussion section.

Response: Thanks for your question and advice. It has been reported that brain was one of the major tissues with distribution of Chr-A in female ICR mice after oral dose of 50 mg/kg of Chr-A, suggesting Chr-A has ability to cross the blood-brain-barrier and enter the brain. Besides, pharmacokinetic parameters evaluation of Chr-A in mice plasma has been performed and reported. [Xu et al. J Nanobiotechnol. Mechanochemical preparation of chrysomycin A self-micelle solid dispersion with improved solubility and enhanced oral bioavailability. (2021) 19:164] We have inserted the relevant point in the Discussion section.

Point 2 Authors showed that Chr-A induces apoptosis of glioma cells both in vivo and in vitro by investigating cleaved-caspase expression levels with western blot technique. to complete better, it would be interesting to evaluate the apoptotic rate in the glioma tissue samples of the control and treated group. In addition, given that data suggest the activation of the mitochondrial pathway, authors should investigate, in vitro, the mitochondria status (for example by using flow cytometry and JC-1 staining) in untreated or Chr-A-treated glioma cells.

Response: Thanks for your advice. We are so sorry that these arguments were not fully taken into account in this study. We will test these valuable indicators according to your suggestion in further research.

Point 3 Figure 1 panel G and Figure 2 especially panels A, B and C are too small. Impossible to appreciate the data. Please improve the size and resolution.

Response: Thanks for your advice. Modifications have been made for Figure 1 and Figure 2 and Figure 2 was separated to Figure 2 and Figure 3.

Point 4 The size of the lettering on figures 3 and 5 is too small. Please improve.

Response: Thanks for your advice. Modifications have been made for Figure 4 and Figure 6.

Reviewer 2 Report

In this manuscript, the authors report the in vivo and in vitro effects of chrysomycin A in glioblastoma xenografts and cell line models. The treatment inhibited the glioblastoma xenograft tumors growth. RNA-seq analysis of the xenografts suggested PI3K/Akt signaling pathway (and apoptosis regulation) be affected, which was further confirmed in treated U87-MG and U251 human glioblastoma cells.

The study is well-designed and the results are mostly well-represented.

Minor corrections and suggestions are as follows:

Line 37: Please provide a context on how GBM, a rare disease but often with a terminal outcome, can be considered „a high social and medical burden“. Reference 5 mentioned the statement in the abstract and background, but didn’t really explore the subject.

I think it would be desirable to provide the context as to why was this paper submitted to the Marine Drugs journal. For example, in the paragraph on lines 42-49, you could mention the origin of the drug.

You claim there are few reports in line 49 - if there are any previous studies with Chr-A on glioblastoma in vivo, please provide references.

The sentence on line 55 should be expanded to a paragraph where you provide information on what you did in the study. I don’t think one sentence covers it.

Please provide a higher-resolution image for Figure 1G.

Please indicate what was used as a blocking agent in 4.3. Ki67 staining by immunofluorescence and for how long. I don’t think the Ki67 antibody is listed where it was purchased and the catalog number, please add the information.

Line 236: please provide a reference to the study where you defined the doses.

Please mark the pathways you decided to explore based on KEGG Pathway and GO Enrichment analysis in Figure 2. I suggest Figures 2D-G be a separate picture, and add in Figure 2 volcano map and heat map of the DEGs.

Other than these, in my opinion, minor corrections, I consider this to be an excellent quality work.

The major criticism of the manuscript is on the account of the English language. I strongly suggest the paper be proofread by a native speaker. In particular, Introduction and Discussion need to be proofread. The expressions are inadequate, despite being vaguely clear (to someone from the area of expertise) what the authors meant to say. For example, I don’t think „strong “ necrosis, „strong“ invasion, and „rapid“ angiogenesis are adequate terms. GBM hasn’t „reached“ the highest level, it is classified as such. The chemotherapeutics are not adopted but approved. Please pay attention to details in expressing yourself.

Please rephrase the sentence on line 209 („Taken our previous...“).

Line 238: ...all the mice were sacrificed...

In Conclusion, please rephrase the „it was apparent“ statement. And you didn’t study the oncogenesis process. „For the next“ is not a complete phrase.

This is just an observation: In some places in the text you wrote Beijing, and in some Peking.

Author Response

Response to Reviewer 2

In this manuscript, the authors report the in vivo and in vitro effects of chrysomycin A in glioblastoma xenografts and cell line models. The treatment inhibited the glioblastoma xenograft tumors growth. RNA-seq analysis of the xenografts suggested PI3K/Akt signaling pathway (and apoptosis regulation) be affected, which was further confirmed in treated U87-MG and U251 human glioblastoma cells. The study is well-designed and the results are mostly well-represented.

Minor corrections and suggestions are as follows:

Point 1 Line 37: Please provide a context on how GBM, a rare disease but often with a terminal outcome, can be considered „a high social and medical burden. Reference 5 mentioned the statement in the abstract and background but didn’t really explore the subject. I think it would be desirable to provide the context as to why was this paper submitted to the Marine Drugs journal. For example, in the paragraph on lines 42-49, you could mention the origin of the drug.

Response: Thanks for your advice. Glioblastoma accounts for 48.6% of malignant brain tumors and it was hard to bring large improvements for old people on prognosis and survival, thus causing a high social and medical burden. [ Miller et al. CA Cancer J Clin. Brain and other central nervous system tumor statistics, 2021. (2021) 71, (5), 381-406.]  Chr-A was derived from Streptomyes sp. 891 from marine sediment, this is consistent with the background of the journal, and we also hope that our study of Chr-A can provides some value evidences to the treatment of glioblastoma, as well as the development prospect of marine-derived drugs. We have added the information in the introduction section.

Point 2 You claim there are few reports in line 49 - if there are any previous studies with Chr-A on glioblastoma in vivo, please provide references.

Response: We are very sorry for the inappropriate use of words. There is no previous report on the effect of Chr-A on glioblastoma in vivo before, and we have corrected this in our manuscript.

Point 3 The sentence on line 55 should be expanded to a paragraph where you provide information on what you did in the study. I don’t think one sentence covers it.

Response: Thanks for your advice. We have added more details about what was done in the study in a new paragraph.

Point 4 Please provide a higher-resolution image for Figure 1G.

Response: Thanks for your advice. Modification has been made for Figure 1G.

Point 5 Please indicate what was used as a blocking agent in 4.3. Ki67 staining by immunofluorescence and for how long. I don’t think the Ki67 antibody is listed where it was purchased and the catalog number, please add the information.

Response: We apologize for not providing comprehensive information about the experimental methods. These informations have been added in the manuscript in 4.3.

Point 6 Line 236: please provide a reference to the study where you defined the doses.

Response: Thanks for your advice. The doses of Chr-A applied to animals was determined by our preliminary experiments, and the concentrations of Chr-A treated on cells was determined by our previous study which has published last year and introduced in our introduction section. The reference has been added in our methods section in 4.6.

Point 7 Please mark the pathways you decided to explore based on KEGG Pathway and GO Enrichment analysis in Figure 2. I suggest Figures 2D-G be a separate picture, and add in Figure 2 volcano map and heat map of the DEGs.

Response: Thanks for your advice. The pathways we explored based on KEGG Pathway and GO Enrichment analysis have been marked in Figure 2. And Figures 2D-G have been separated to a new picture (Figure 3). Besides, Volcano map and Heat map of the DEGs were added in Figure 2.

Point 8 The major criticism of the manuscript is on the account of the English language. I strongly suggest the paper be proofread by a native speaker. In particular, Introduction and Discussion need to be proofread. The expressions are inadequate, despite being vaguely clear (to someone from the area of expertise) what the authors meant to say. For example, I don’t think „strong “necrosis, „strong“ invasion, and „rapid“ angiogenesis are adequate terms. GBM hasn’t reached the highest level, it is classified as such. The chemotherapeutics are not adopted but approved. Please pay attention to details in expressing yourself.

Response: Thanks for your advice. Corrections have been made in the section.

Point 9 Please rephrase the sentence on line 209 („Taken our previous...“).

Response: Thanks for your advice. We have corrected it in the manuscript.

Point 10 Line 238: ...all the mice were sacrificed...

Response: Thanks for your advice. We have corrected it in the manuscript.

Point 11 In Conclusion, please rephrase the „it was apparent“ statement. And you didn’t study the oncogenesis process. „For the next“ is not a complete phrase.

Response: Thanks for your advice. We have corrected it in the manuscript.

Point 12 This is just an observation: In some places in the text you wrote Beijing, and in some Peking.

Response: Thanks for your question. Peking Union Medical College is a proper noun, and the current place name is Beijing.

Reviewer 3 Report

The manuscript entitled "Chrysomycin A regulates proliferation and apoptosis of neuroglioma cells via Akt/GSK-3β signaling pathway in vivo and in vitro" by Dong-Ni Liu et al considers a compound with known activity on tumor cells of inhibition on proliferation, migration and invasion. The authors state that they would like to investigate molecular mechanisms but in the final I cannot see any specific mechanism. The authors should have said that they have already studied this experimental model "in vitro" in a previous published paper (doi: 10.3390/molecules27196148) and that in this study they were trying to evaluate its molecular mechanisms.  In this study the authors make a whole analysis on databases without any meaning and do not really identify a specific pathway. Figure 2B does not read well. The cytograms are all out of scale!. Not publishable. Also The Akt pathway is deregulated in most cancers as well as anti-cancer drugs induce apoptosis and the mechanism by which they pass has not been investigated. The paper is also completely out of the context and scope of the journal as no new compound of marine origin has been discovered or investigated.

Author Response

Response to Reviewer 3

Point 1 The manuscript entitled "Chrysomycin A regulates proliferation and apoptosis of neuroglioma cells via Akt/GSK-3β signaling pathway in vivo and in vitro" by Dong-Ni Liu et al considers a compound with known activity on tumor cells of inhibition on proliferation, migration and invasion. The authors state that they would like to investigate molecular mechanisms but in the final I cannot see any specific mechanism. The authors should have said that they have already studied this experimental model "in vitro" in a previous published paper (doi: 10.3390/molecules27196148) and that in this study they were trying to evaluate its molecular mechanisms.

Response: Thanks for your advice. We are sorry for the lack of an appropriate representation of the purpose of the study and we have made clear the significance and purpose of this research in the introduction section combined with your suggestions.

Point 2 In this study the authors make a whole analysis on databases without any meaning and do not really identify a specific pathway.

Response: Thanks for your comment. We are sorry that our manuscript gives you such an opinion on this study. To better elucidate our study, we added volcano map and heat map of the DEGs in Figure 2 and the pathways we explored based on KEGG Pathway and GO Enrichment analysis were marked in Figure 2 which played an important role in our study indicating that Chr-A may regulate apoptosis, PI3K/Akt and Wnt signaling pathway. In our previous published paper, Chr-A inhibit proliferation, migration and invasion of neuroglioma cells via Akt/GSK-3β signal in vitro. Corresponding to the enrichment of DEGs in this study, Akt/GSK-3β signal builds a bridge for the interaction of PI3K/Akt and Wnt signaling pathway in cancers. Thus, we hope to validate the regulation of Chr-A on Akt/GSK-3β signaling pathway in vivo here.

Point 3 Figure 2B does not read well. The cytograms are all out of scale!

Response: Thanks for your advice. We improve the size of Figure1G to make it clear. We hope that such processing and explanation can eliminate your doubts. If you still have any questions or suggestions for modification, we sincerely hope to hear from you again.

Point 4 The Akt pathway is deregulated in most cancers as well as anti-cancer drugs induce apoptosis and the mechanism by which they pass has not been investigated. The paper is also completely out of the context and scope of the journal as no new compound of marine origin has been discovered or investigated.

Response: Thanks for your comment. The Akt signaling pathway is very important in GBM. Humans have three AKT isoforms: AKT1, AKT2, and AKT3. Studies showed that the expression of Akt1 protein and mRNA was similar in glioma and normal control tissues, while the protein and mRNA level of Akt2 increased with the pathological grade of malignancy, the expression of Akt3 mRNA and protein decreased as the malignancy grade increased. Interestingly, knockdown of AKT2 or AKT3 in glioma cell lines suppressed colony formation and caspase-induced apoptosis [Hideo et al. Neuro Oncol. Akt2 and Akt3 play a pivotal role in malignant gliomas. 2010 Mar;12(3):221-32]. Upregulation of the PI3K/Akt/mTOR signaling pathway is closely-related with glioblastoma to mediate metabolism reprogramming, cytoskeletal rearrangement, metabolism, apoptosis, and angiogenesis. [Elena et al. Cancer Commun (Lond). An update on the molecular biology of glioblastoma, with clinical implications and progress in its treatment. 2022 Nov;42(11):1083-1111. Zhu et al., Front Pharmacol. Celastrol Suppresses Glioma Vasculogenic Mimicry Formation and Angiogenesis by Blocking the PI3K/Akt/mTOR Signaling Pathway. 2020, 11, 25.2930]. Sinomenine ester derivative inhibits glioblastoma with the reduced expression of p-AKT/AKT in U87 and U251 cells [Elena et al. Cancer Commun (Lond). An update on the molecular biology of glioblastoma, with clinical implications and progress in its treatment. 2022 Nov;42(11):1083-1111.]. A-443654, a Akt inhibitor, inhibits growth and induces apoptosis of glioblastoma cells in vitro [Gary L et al. Mol Cancer Ther. Inhibition of Akt inhibits growth of glioblastoma and glioblastoma stem-like cells. 2009 Feb;8(2):386-93.]. Perifosine, another Akt inhibitor, could inhibit proliferation of Ovarian cancer cells. [Entidhar et al. Gynecol Oncol. Perifosine, an AKT inhibitor, modulates ovarian cancer cell line sensitivity to cisplatin-induced growth arrest. 2013 Oct;131(1):207-12.]

Point 5 The paper is also completely out of the context and scope of the journal as no new compound of marine origin has been discovered or investigated.

Response: Thanks for your comment. Chr-A was derived from Streptomyes sp. 891 from marine sediment [Ni et al. Prep Biochem Biotechnol. Optimization of fermentation conditions and medium compositions for the production of chrysomycin a by a marine-derived strain Streptomyces sp. 891. 2021, 51, (10), 998-1003.], this is consistent with the context and scope of the journal. Although Chr-A was reported to have anti-tumor activity before, the effect and mechanisms of Chr-A on GBM is unknown. Thus, we hope that our study of Chr-A could be accepted by Marine Drugs and could provide some value evidences to the treatment of glioblastoma, as well as the development prospect of marine-derived drugs.

Round 2

Reviewer 3 Report

I greatly appreciate the manuscript authors' response, which is convincing in several respects. However, I must point out that the cytograms shown are not correct and therefore the tool needs to be set up so that the controls fall in the reference domain. In my opinion it is not possible to publish cytograms as shown by the authors.

Author Response

Point: I greatly appreciate the manuscript authors' response, which is convincing in several respects. However, I must point out that the cytograms shown are not correct and therefore the tool needs to be set up so that the controls fall in the reference domain. In my opinion it is not possible to publish cytograms as shown by the authors.

Response: Thank you very much for your suggestion again. We are very sorry that we failed to correct the Figure you mentioned last time. Now Figure 2 has been corrected (marinedrugs-2377544-R2).

Round 3

Reviewer 3 Report

My comment regarding cytograms are referring to Figure 4. If you compare your cytograms with other published cytograms you can see that the controls must be calibrated within a certain quadrant or section and then evaluate the shifting of the other samples. If the cytofluorometer protocol is not calibrated correctly it is very difficult to evaluate the data correctly. In my opinion it is necessary to change the cytograms before the paper is published.

Author Response

Point:My comment regarding cytograms are referring to Figure 4. If you compare your cytograms with other published cytograms you can see that the controls must be calibrated within a certain quadrant or section and then evaluate the shifting of the other samples. If the cytofluorometer protocol is not calibrated correctly it is very difficult to evaluate the data correctly. In my opinion it is necessary to change the cytograms before the paper is published.

Response: Thank you for pointing out this critical issue. The data of apoptosis rate and CAS 3/7 activity of U251 and U87 cells after treatment with Chr-A has been re-checked and processed, and Fig4 has been replotted (marinedrugs-2377544-R3).

Round 4

Reviewer 3 Report

Accept in present form